# Illness severity and risk of mental morbidities among patients recovering from COVID-19: a cross-sectional study in the Icelandic population

Karen Sól Saevarsdóttir,[1] Hildur Ýr Hilmarsdóttir ![ORCID],[1] Ingibjörg Magnúsdóttir,[1] Arna Hauksdóttir,[1] Edda Bjork Thordardottir,[1] Ásdís Braga Gudjónsdóttir,[1] Gunnar Tomasson,[1,2] Harpa Rúnarsdóttir,[1] Harpa Lind Jónsdóttir,[1,3] Berglind Gudmundsdóttir,[4] Gudrún Pétursdóttir,[5] Pétur Henry Petersen,[4] Sigurdur Yngvi Kristinsson,[2,4] Thorvardur Jon Love,[2,4] Sif Hansdóttir,[2] Hrönn Hardardóttir,[2] Gunnar Gudmundsson,[2,4] Elias Eythorsson,[2] Dóra Gudrún Gudmundsdóttir,[6] Hildur Sigbjörnsdóttir,[6] Sigrídur Haraldsdóttir,[6] Alma Dagbjört Möller,[6] Runolfur Palsson,[2,4] Jóhanna Jakobsdóttir,[1] Thor Aspelund ![ORCID],[1,7] Unnur Valdimarsdottir ![ORCID] [1,8,9]

KSS, HÝH, IM, JJ, TA and UV contributed equally.

**Correspondence to**
Dr Unnur Valdimarsdottir;
unnurav@hi.is

## ABSTRACT

**Objective** To test if patients recovering from COVID-19 are at increased risk of mental morbidities and to what extent such risk is exacerbated by illness severity.

**Design** Population-based cross-sectional study.

**Setting** Iceland.

**Participants** A total of 22 861 individuals were recruited through invitations to existing nationwide cohorts and a social media campaign from 24 April to 22 July 2020, of which 373 were patients recovering from COVID-19.

**Main outcome measures** Symptoms of depression (Patient Health Questionnaire), anxiety (General Anxiety Disorder Scale) and posttraumatic stress disorder (PTSD; modified Primary Care PTSD Screen for DSM-5) above screening thresholds. Adjusting for multiple covariates and comorbidities, multivariable Poisson regression was used to assess the association between COVID-19 severity and mental morbidities.

**Results** Compared with individuals without a diagnosis of COVID-19, patients recovering from COVID-19 had increased risk of depression (22.1% vs 16.2%; adjusted relative risk (aRR) 1.48, 95% CI 1.20 to 1.82) and PTSD (19.5% vs 15.6%; aRR 1.38, 95% CI 1.09 to 1.75) but not anxiety (13.1% vs 11.3%; aRR 1.24, 95% CI 0.93 to 1.64). Elevated relative risks were limited to patients recovering from COVID-19 that were 40 years or older and were particularly high among individuals with university education. Among patients recovering from COVID-19, symptoms of depression were particularly common among those in the highest, compared with the lowest tertile of influenza-like symptom burden (47.1% vs 5.8%; aRR 6.42, 95% CI 2.77 to 14.87), among patients confined to bed for 7 days or longer compared with those never confined to bed (33.3% vs 10.9%; aRR 3.67, 95% CI 1.97 to 6.86) and among patients hospitalised for COVID-19 compared with those never admitted to hospital (48.1% vs 19.9%; aRR 2.72, 95% CI 1.67 to 4.44).

**Conclusions** Severe disease course is associated with increased risk of depression and PTSD among patients recovering from COVID-19.

## Strengths and limitations of this study

► To our knowledge, this is the first study addressing mental morbidities among recovering COVID-19 patients in a general population setting, including outpatients.

► This study includes more than 20% of all patients diagnosed with COVID-19 by reverse transcription-PCR during the first wave of the epidemic in Iceland, where the diagnostic rate was among the highest worldwide, allowing us to perform sensitivity analyses limited to individuals who all had been screened for SARS-CoV-2.

► The extensive and well-piloted questionnaire allowed us to obtain validated symptom assessment of mental morbidities along with detailed information on influenza-like symptom burden for all individuals during the influenza season in Iceland.

► The cross-sectional design is less informative on the directionality of the studied associations, and as the study is nested within a small, economically and socially secure society, the generalisability of the findings may be limited.

► It is possible that individuals with severe symptoms of depression and anxiety were more likely to report high levels of influenza-like symptoms during the preceding 2 months when responding to the questionnaire, yielding an outcome-dependent misclassification.

## INTRODUCTION

With almost 180 million reported cases and 3.9 million deaths globally,[1] the COVID-19 pandemic remains a major public health challenge worldwide. Beyond the extensive socioeconomic impact and suffering of patients during

the illness, there are rising concerns about long-term somatic and psychological impact on recovering patients.[2–6]

Mental health consequences among patients in recent epidemics are well documented.[7 8] With respect to COVID-19, a meta-analysis of 31 studies suggested a 45%–47% pooled prevalence of mild to severe symptoms of depression and anxiety in patients with COVID-19, mostly inpatients.[9] Moreover, extreme rates of posttraumatic stress have been reported among COVID-19 inpatients in China.[10 11] Previous studies are often small, limited to inpatients and some lack a control group. Fewer studies have reported symptoms of mental morbidities after recovery from COVID-19, with existing data suggesting high levels of symptoms[12] and elevated rates of diagnosed psychiatric disorders among COVID-19 inpatients during the first weeks and months after hospital discharge.[13–15]

Influenza-like symptoms, such as myalgia, cough and sore throat, have been associated with mental morbidities in the general population during the COVID-19 pandemic.[16 17] To what extent influenza-like symptoms are associated with psychological distress in patients recently recovered from COVID-19 remains unknown. There are several mechanisms through which COVID-19 may be associated with mental morbidity. First, uncertainty regarding the disease course may give rise to symptoms of anxiety and depression in infected patients.[18] Second, serious infection may induce a cytokine storm,[19 20] leading to exacerbation of the illness and development of psychological symptoms such as depression.[19] The reverse may also be true that psychological symptoms may cause more severe COVID-19 illness through excessive or dysregulated inflammation.[21]

The first wave of COVID-19 in Iceland started on 28 February 2020. Through modest but systematic mitigation strategies,[22] the incidence of COVID-19 had effectively been attenuated by the end of April 2020, with only sporadic cases occurring until the onset of the second wave of infections on 22 July. During this period, more than 20% of the total population was tested using reverse transcription-PCR (RT-PCR); a total of 1820 tested positive for SARS-CoV-2 (0.5% of the total population), of whom 113 were hospitalised (6%) and 10 died (0.5%).[23] Here, we explore mental health indicators in patients recovering from COVID-19 in the first wave of the pandemic and the potential role of disease severity on the risk of mental morbidities.

## METHODS
### Study population and design
The COVID-19 National Resilience Cohort was established on 24 April 2020, with the overarching aim of understanding the long-term public health impact of the COVID-19 epidemic in Iceland.[24] Eligible were all Icelandic and English-speaking individuals 18 years or older who had an Icelandic electronic ID (total population ≥18 years on 1 January 2020 was 282 770[25]). Recruitment was mainly through social media and public media coverage, although participants in ongoing cohort studies (The SAGA (Stress-And-Gene-Analysis)

cohort (N=31 795 women); iStopMM (N=80 730 men and women) and Health and well-being of Icelanders (N=6102 men and women)) received email or postal invitations to participate. In addition, all individuals in Iceland who tested positive by RT-PCR for SARS-CoV-2 through May 2020 (N=1800) received a text message invitation on 3 June 2020. All participants signed an electronic informed consent and subsequently answered a web-based questionnaire. The study was approved by the National Bioethics Committee (number 20-073) and the Data Protection Authority.

We performed a cross-sectional analysis of participants responding to the baseline questionnaire from 24 April through 22 July 2020. A total of 22 680 individuals had participated at that time point. We excluded individuals who did not state whether or not they had undergone a diagnostic test (n=989) or been diagnosed with COVID-19 (n=19) and those who did not answer all questions on influenza-like symptoms (n=762) or all items in any one of the three measures of depression, anxiety and posttraumatic stress (n=49) (see online supplemental figure 1). The final analytic sample consisted of 20 861 participants.

### Measures
#### Sociodemographic and health-related factors
Sociodemographic information included the date of response to the baseline questionnaire, gender, age, education, average monthly income during the past year, relationship status and residency (capital vs other regions). We defined relationship status as either being in a relationship (married, cohabiting or in a relationship yet not cohabiting) or not (single, widowed or divorced). Highest level of education was defined as (1) compulsory, (2) upper secondary/vocational/other, (3) university examination (bachelor's degree or diploma certificate) or (4) master's or doctoral degree. Monthly income categories were defined as (1) low income, <1653 GBP (British pound sterling); (2) low–medium income, 1653–2755 GBP; (3) medium income, 2756–3858 GBP; (4) medium–high income, 3859–5511 GBP; (5) high income, >5511 GBP (conversion rates according to The Central Bank of Iceland, 24 April 2020[26]).

Health-related information included current smoking status (yes/no), binge drinking during the past 2 months (defined as ≥4 drinks for women and ≥5 drinks for men or other genders[27]), a previous diagnosis of a psychiatric disorder (made by physicians or psychologists; yes/ no) and previous diagnosis of somatic comorbidities including hypertension, diabetes, heart disease, lung disease, chronic kidney disease, cancer and immunosuppressive state or immunosuppressive therapy (none, one or two or more comorbid conditions). Additionally, participants reported their height (m) and weight (kg), which was used to estimate body mass index as kg/m$^2$.

#### Diagnosis of COVID-19 and illness severity
We asked participants whether they had been tested for COVID-19 regardless of testing method. If they

responded yes, we asked whether they had been diagnosed with COVID-19, and if so, how long ago they had been diagnosed, how long they were confined to bed due to COVID-19, and whether they had been hospitalised.

As the first wave of COVID-19 coincided with the influenza season in Iceland, we asked all participants how many days they had experienced nine specific influenza-like symptoms over the preceding 2 months. The symptoms, previously identified as relevant for COVID-19,[28][29] were fever, cough, sore throat, impaired sense of taste or smell, headache, myalgia (body aches), gastrointestinal symptoms (abdominal pain, nausea, vomiting, diarrhoea), shortness of breath, fatigue and weakness. Response options were 'never', '1–2 days', '3–6 days', '1–2 weeks' and 'more than 2 weeks', which were scored from '0' (never) to '4' (≥2 weeks). We calculated influenza-like symptom burden by summing all symptom scores indicated by each participant (range 0–36 points) and then divided scores into tertiles of the distribution within the COVID-19 cohort (lowest 0–11, mid 12–23, highest 24–36 points).

### Symptoms of mental illness

We used the 9-item Patient Health Questionnaire (PHQ-9) to measure symptoms of depression with the recommended cut-off of ≥10 points serving as a screening indicator for depression in an adult primary care sample.[30] We used the 7-item Generalised Anxiety Disorder scale (GAD-7) to assess symptoms of anxiety. A cut-off of ≥10 was employed as an indicator of moderate to severe symptoms of a generalised anxiety disorder in an adult primary care sample.[31] The 5-item Primary Care PTSD Screen for DSM-5 (PC-PTSD-5) was used to measure symptoms of posttraumatic stress,[32] with a modified version tailored to COVID-19 (eg, 'Had nightmares about COVID-19?'). We scored the responses as '0' (never) or '1' (seldom, sometimes, often, very often) with a total score ranging from 0 to 5. We defined COVID-19-specific posttraumatic stress disorder (PTSD) as a PC-PTSD-DSM-5 score of ≥4, which has previously been reported as optimal.[32]

### Statistical analysis

First, we contrasted the distribution of sociodemographic and health-related factors between individuals with and without a COVID-19 diagnosis. We then ran univariable and multivariable linear regression to detect potential differences in continuous symptoms of depression (PHQ-9), anxiety (GAD-7) and PTSD (PC-PTSD-5) across groups with and without a confirmed COVID-19 diagnosis. We used robust (modified) Poisson regression, with the classical sandwich estimator,[33] to determine the association between a confirmed diagnosis of COVID-19 and risk of depression, anxiety or PTSD surpassing cut-offs of potential clinical significance. The association is presented as unadjusted and adjusted relative risks (aRRs) with 95% CIs and as prevalence differences. All multivariable models included age (continuous), gender (male, female or other), educational level (classification described

above), income (continuous), current smoking (yes/no), previous diagnosis of a psychiatric disorder (yes/no), number of previously diagnosed somatic diseases (0, 1 or ≥2) and the time period of responding to the baseline questionnaire (24–30 April, 1–7 May, 8–23 May or 24 May to 22 July). We then performed these analyses stratified by all covariates and performed a likelihood ratio test for effect modification.

We performed several sensitivity analyses. As we were concerned for a potential overlap between physical symptoms of COVID-19 and some items on the mental health assessments (eg, fatigue), we performed a sensitivity analysis excluding patients diagnosed with COVID-19 within 14 days of responding to the questionnaire. We further reran the analysis using individuals tested, but not positive for COVID-19 as a control group. There was a considerable difference between individuals with and without a COVID-19 diagnosis with respect to the date of responding to the questionnaire (table 1). Therefore, we repeated the multivariable Poisson regression stratified by questionnaire answer date, divided into two groups (24 April to 7 May and 8 May to 22 July). Finally, as our primary Poisson regression analysis included individuals with complete responses to PHQ-9 (n=18 822), GAD-7 (n=19 163) and PC-PTSD-5 (n=17 562), we repeated these analyses, including the entire analytic sample (n=20 861), using multiple imputation by creating 10 imputed data sets with 10 iterations each.[34]

Limited to individuals diagnosed with COVID-19, we used multivariable Poisson regression to evaluate the association between indicators of COVID-19 severity and mental morbidities. Adjusting for the same covariates as listed above, we explored the association between time confined to bed due to COVID-19 (never vs 1–6 days or 7 days or more), influenza-like symptom burden (in tertiles) and whether or not patients had been hospitalised, and mental morbidities among recovering patients.

Next, we compared the proportion of individuals with and without a COVID-19 diagnosis (tested and not tested for SARS-CoV-2) who reported each of the nine influenza-like symptoms for 1 week or more. We used robust Poisson regression to calculate unadjusted and multivariable-aRRs of each symptom among individuals tested versus not tested for SARS-CoV-2 and then diagnosed versus not diagnosed with COVID-19. The multivariable models included the same covariates as listed above. Using linear regression, we then tested the association between influenza-like symptom burden and mental morbidities both among individuals with and without a diagnosis of COVID-19 (tested and not tested for SARS-CoV-2). We then tested whether influenza-like symptoms mediated the association between a confirmed COVID-19 diagnosis and mental morbidities by adding influenza-like symptoms to the multivariable models. All analyses were conducted in R (V.3.6.2).

**Table 1** Characteristics of the study population, with and without a diagnosis of COVID-19

| | Number (%) | | |
| --- | --- | --- | --- |
| | Individuals not diagnosed with COVID-19 (n=20 488) | Individuals diagnosed with COVID-19 (n=373) | P value |
| **Gender** | | | |
| Male | 6133 (29.9) | 121 (32.4) | 0.15 |
| Female | 14 306 (69.8) | 250 (67.0) | |
| Other | 36 (0.2) | 2 (0.5) | |
| Missing | 13 (0.1) | 0 (0) | |
| **Age** | | | |
| 18–29 years | 1191 (5.8) | 51 (13.7) | <0.001 |
| 30–39 years | 1949 (9.5) | 39 (10.5) | |
| 40–49 years | 3644 (17.8) | 99 (26.5) | |
| 50–59 years | 5309 (25.9) | 99 (26.5) | |
| 60–69 years | 5359 (26.2) | 74 (19.8) | |
| 70 years or older | 3036 (14.8) | 11 (2.9) | |
| **Highest educational level** | | | |
| Compulsory education | 2865 (14.0) | 30 (8.0) | 0.008 |
| Upper secondary, vocational or other education | 6303 (30.8) | 117 (31.4) | |
| Bachelor's degree or diploma certificate | 6517 (31.8) | 135 (36.2) | |
| Master's or PhD degree | 4685 (22.9) | 91 (24.4) | |
| Missing | 118 (0.6) | 0 (0) | |
| **Average monthly income*** | | | |
| Low income | 3527 (17.2) | 50 (13.4) | <0.001 |
| Low-medium income | 5642 (27.5) | 88 (23.6) | |
| Medium income | 4993 (24.4) | 82 (22.0) | |
| Medium-high income | 3527 (17.2) | 91 (24.4) | |
| High income | 1825 (8.9) | 47 (12.6) | |
| Missing | 974 (4.8) | 15 (4.0) | |
| **Residence** | | | |
| Capital area | 13 986 (68.3) | 270 (72.4) | 0.18 |
| Elsewhere in Iceland | 6352 (31.0) | 102 (27.3) | |
| Abroad | 140 (0.7) | 1 (0.3) | |
| Missing | 10 (0.0) | 0 (0) | |
| **Marital status** | | | |
| In a relationship | 15 735 (76.8) | 302 (81.0) | 0.09 |
| Single | 4671 (22.8) | 71 (19.0) | |
| Missing | 82 (0.4) | 0 (0) | |
| **BMI category (kg/m²)** | | | |
| <25, normal weight | 5931 (28.9) | 112 (30.0) | 0.70 |
| 25–30, overweight | 7931 (38.7) | 138 (37.0) | |
| >30, obese | 6127 (29.9) | 118 (31.6) | |
| Missing | 499 (2.4) | 5 (1.3) | |
| **Smoking status** | | | |
| No | 17 873 (87.2) | 345 (92.5) | 0.002 |
| Yes | 2521 (12.3) | 26 (7.0) | |
| Missing | 94 (0.5) | 2 (0.5) | |

Continued

**Table 1** Continued

| | Number (%) | | |
| --- | --- | --- | --- |
| | Individuals not diagnosed with COVID-19 (n=20 488) | Individuals diagnosed with COVID-19 (n=373) | P value |
| **Binge drinking** | | | |
| No | 18 255 (89.1) | 312 (83.6) | 0.02 |
| Yes | 2233 (10.9) | 61 (16.4) | |
| **Previous diagnosis of psychiatric disorder** | | | |
| No | 14 510 (70.8) | 285 (76.4) | 0.02 |
| Yes | 5776 (28.2) | 84 (22.5) | |
| Missing | 202 (1.0) | 4 (1.1) | |
| **Physical diseases** | | | |
| No comorbidities | 11 958 (58.4) | 263 (70.5) | <0.001 |
| One comorbidity | 5964 (29.1) | 86 (23.1) | |
| Two or more comorbidities | 2453 (12.0) | 21 (5.6) | |
| Missing | 113 (0.6) | 3 (0.8) | |
| **Questionnaire answer date** | | | |
| 24–30 April | 7841 (38.3) | 102 (27.3) | <0.001 |
| 1–7 May | 5413 (26.4) | 76 (20.4) | |
| 8–23 May | 4999 (24.4) | 53 (14.2) | |
| 24 May–22 July | 2235 (10.9) | 142 (38.1) | |
| **Time since diagnosis** | | | |
| Less than 2 weeks | – | 8 (2.1) | – |
| 2–4 weeks | – | 60 (16.1) | |
| More than 4 weeks | – | 305 (81.8) | |

*Income categories were defined as: low income, <1653 GBP; low-medium income, 1653–2755 GBP; medium income, 2756–3858 GBP; medium-high income, 3859–5511 GBP; high income, >5511 GBP (conversion rates according to Central Bank of Iceland, 24 April 2020).

## Patient and public involvement

No patients were involved in putting forward the research question or the outcome measures, nor were they involved in developing plans for design or implementation of the study. Dissemination of the results to study participants and the Icelandic population will be obtained through a media outreach (eg, press release and communication on our study website) on publication of this study.

## RESULTS
### Background characteristics

Of the 20 861 participants, 5419 individuals had been tested for SARS-CoV-2, of whom 373 reported having been diagnosed with COVID-19 (97.9% more than 2 weeks before responding to the questionnaire; table 1). Compared with other participants, individuals previously diagnosed with COVID-19 were younger (mean age: 48.3 vs 54.8 years), had a higher educational level and income and were more likely to binge drink alcohol. They were furthermore less likely to be current smokers and have

**Table 2** The prevalence, crude and multivariable adjusted relative risks and absolute differences in symptoms of depression (PHQ-9), anxiety (GAD-7) and PTSD (PC-PTSD-5) surpassing screening thresholds among individuals with and without a confirmed diagnosis of COVID-19

| | Number (%) | | cRR (95% CI) | aRR (95% CI)* | Absolute difference crude % | Absolute difference adjusted* % |
|---|---|---|---|---|---|---|
| | Individuals not diagnosed with COVID-19 | Individuals diagnosed with COVID-19 | | | | |
| Depression | 2992 (16.2) | 75 (22.1) | 1.36 (1.11 to 1.67) | 1.48 (1.20 to 1.82) | 5.9 | 8.4 |
| Anxiety | 2120 (11.3) | 45 (13.1) | 1.16 (0.88 to 1.53) | 1.24 (0.93 to 1.64) | 1.8 | 2.6 |
| PTSD | 2699 (15.6) | 59 (19.5) | 1.25 (0.99 to 1.57) | 1.38 (1.09 to 1.75) | 3.9 | 7.2 |

*Adjusted for age (continuous variable), gender (male, female, or other), educational level (compulsory education, high school/trade school/other education, bachelor's degree/diploma certificate, or master's/PhD degree), income (continuous variable), current smoking (yes or no), previous diagnosis of a psychiatric disorder (yes or no), number of previously diagnosed somatic diseases (0, 1, or ≥2), and timing of responding to the baseline questionnaire (24–30 April, 1–7 May, 8–23 May or 24 May–22 July).
aRR, adjusted relative risk; cRR, crude relative risk; GAD, General Anxiety Disorder; PC-PTSD-5, Primary Care Posttraumatic Stress Disorder Screen for DSM-5; PHQ, Patient Health Questionnaire.

previously been diagnosed with psychiatric disorders or somatic diseases. The analytic sample had similar education[35] and residence distribution as the general population, while the median age was higher and women were over-represented.[36]

## COVID-19 and mental morbidities

Individuals diagnosed with COVID-19 reported, compared with others, higher mean scores of depression (6.11 vs 5.07; p<0.001) and PTSD (1.90 vs 1.72; p<0.001), but not anxiety (4.52 vs 4.12; p=0.08; see online supplemental table 1). Similarly, multivariable-aRRs of depression (22.1% vs 16.2%; aRR 1.48, 95% CI 1.20 to 1.82) and PTSD (19.5% vs 15.6%; aRR 1.38, 95% CI 1.09 to 1.75) above symptom thresholds were increased among those recovering from COVID-19 compared with others, while not on anxiety (13.1% vs 11.3%; aRR 1.24, 95% CI 0.93 to 1.64; table 2). Risk elevations in mental morbidity in individuals diagnosed with COVID-19 were limited to those 40 years or older and was highest among individuals 60 years or older (see online supplemental table 2). Individuals with higher educational level were more likely to suffer from symptoms of depression and anxiety after a diagnosis of COVID-19. When stratified by answer date, risk elevations for depression and anxiety in patients recovering from COVID-19 were lower for those who answered in the later response period (ie, 8 May to 22 July), whereas it was higher for PTSD (see online supplemental table 2) and no statistically significant interactions were observed. Limiting the analysis to individuals who were diagnosed with COVID-19 more than 2 weeks before responding to the questionnaire and repeating the analysis using multiple imputation yielded virtually identical results (see online supplemental tables 3,4). Finally, limiting the analysis to individuals who were tested for COVID-19 generated a similar pattern yet slightly lower point estimate (see online supplemental table 5).

## COVID-19 illness severity and mental morbidities

In table 3, we show the prevalence and aRRs of mental morbidities among individuals with a diagnosis of COVID-19 by number of days confined to bed, hospitalisation and severity of COVID-19 symptoms. We observed dose-dependent associations between these indices of COVID-19 severity and risk of mental morbidities. Risk elevations by days confined to bed and hospitalisation due to COVID-19 were statistically significant for depression and anxiety, while influenza-like symptom burden was associated with all measures of mental morbidity.

## Influenza-like symptoms and mental morbidities

In figure 1, we present the proportions of individuals with influenza-like symptoms lasting at least 1 week during the preceding 2 months among patients recovering from COVID-19, those who tested negative and those never tested for SARS-CoV-2 (more detailed in online supplemental table 6). Individuals who tested negative reported 29%–91% higher prevalence of symptoms than those never tested for SARS-CoV-2 (see online supplemental table 7), while individuals with a confirmed COVID-19 diagnosis had, compared with all others, dramatically increased adjusted risk ratios of all symptoms: 3.49 for gastrointestinal symptoms, 3.66 for myalgia, 4.16 for fatigue, 4.58 for sore throat, 4.64 for headache, 9.26 for shortness of breath, 19.10 for fever and 32.52 for impaired sense of taste or smell (see online supplemental table 8).

We found that influenza-like symptom burden was positively associated with symptom levels of depression, anxiety and PTSD, regardless of COVID-19 diagnosis (figure 2, online supplemental table 9). A stepwise increase across tertiles in influenza-like symptom burden was associated with a rise in mean levels of depression, anxiety and PTSD among patients recovering from COVID-19, individuals who tested negative and those never tested for SARS-CoV-2. Patients recovering from COVID-19 reporting low to medium influenza-like symptom burden (low and mid tertiles) presented with lower levels of depression and anxiety compared with those with same levels of influenza-like symptom burden but without a diagnosis of COVID-19 (figure 2, online supplemental table 9).

Finally, when influenza-like symptom burden was added to the multivariable models (presented in table 2), the direction

**Table 3** Prevalence and adjusted relative risks (95% CI) of mental morbidities among patients recently recovering from COVID-19 by disease and symptom severity

| | | Depression | | Anxiety | | PTSD | |
|---|---|---|---|---|---|---|---|
| | n | %≥10 PHQ-9 | aRR* (95% CI) | %≥10 GAD-7 | aRR* (95% CI) | %≥4 PC-PTSD-5 | aRR* (95% CI) |
| Confined to bed due to COVID-19 | | | | | | | |
| Never | 136 | 10.9 | Ref. | 9.9 | Ref. | 15.6 | Ref. |
| 1–6 days | 131 | 24.2 | 2.12 (1.11 to 4.02) | 11.3 | 1.13 (0.55 to 2.32) | 18.2 | 0.94 (0.53 to 1.69) |
| 7 days or more | 105 | 33.3 | 3.67 (1.97 to 6.86) | 19.4 | 2.58 (1.29 to 5.15) | 26.2 | 1.65 (0.92 to 2.96) |
| Hospitalised for COVID-19 | | | | | | | |
| No | 341 | 19.9 | Ref. | 12.7 | Ref. | 18.3 | Ref. |
| Yes | 32 | 48.1 | 2.72 (1.67 to 4.44) | 17.2 | 1.74 (0.85 to 3.57) | 32.0 | 2.05 (0.97 to 4.32) |
| COVID-19 symptom severity | | | | | | | |
| Lowest tertile | 115 | 5.8 | Ref. | 4.9 | Ref. | 9.6 | Ref. |
| Mid tertile | 161 | 18.8 | 2.70 (1.16 to 6.28) | 12.1 | 2.15 (0.81 to 5.70) | 20.1 | 1.76 (0.88 to 3.55) |
| Highest tertile | 97 | 47.1 | 6.42 (2.77 to 14.87) | 23.9 | 3.91 (1.45 to 10.51) | 30.7 | 2.70 (1.30 to 5.59) |

*Adjusted for age (continuous variable), gender (male, female, or other), educational level (compulsory education, high school/trade school/other education, bachelor's degree/diploma certificate, or master's/PhD degree), income (continuous variable), current smoking (yes or no), previous diagnosis of a psychiatric disorder (yes or no), number of previously diagnosed somatic diseases (0, 1 or ≥2), and timing of responding to the baseline questionnaire (24–30 April, 1–7 May, 8–23 May, or 24 May–22 July).
aRR, adjusted relative risk; cRR, crude relative risk; GAD, General Anxiety Disorder; PC-PTSD-5, Primary Care Posttraumatic Stress Disorder Screen for DSM-5; PHQ, Patient Health Questionnaire.

of the aRRs of depression (0.63, 95% CI 0.51 to 0.77) and PTSD (0.73, 95% CI 0.57 to 0.93) reversed in patients recovering from COVID-19 compared with others, indicating that influenza-like symptoms mediated the risks of mental morbidity in this otherwise prepandemic healthy population.

## DISCUSSION

The findings of this study suggest that patients recovering from COVID-19 may experience elevated risks of depression and PTSD, particularly if recovering from a severe disease. We found that mental morbidities among patients recovering from COVID-19 were strongly associated with

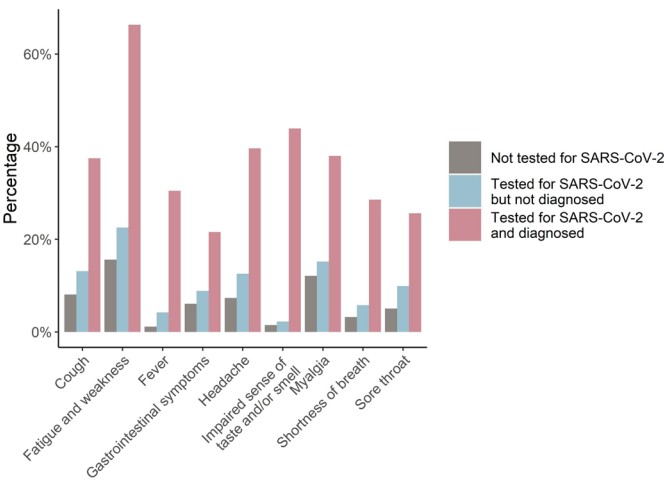

**Figure 1** The proportion of individuals with and without (tested and not tested) a confirmed diagnosis of COVID-19 with influenza-like symptoms of 1 week or more during the last 2 months before responding to the baseline questionnaire.

older age, higher educational level, greater influenza-like symptom burden, extended time confined to bed and hospitalisation due to COVID-19.

To our knowledge, this is the first study addressing mental morbidities among patients recovering from COVID-19 in a general population setting, including outpatients. Our findings are in line with the vast literature reporting high rates of mental morbidities among current or discharged COVID-19 inpatients.[9 10 12 13] Several mechanisms may play a role in the increased risk of mental health morbidities among patients recovering from a severe COVID-19 illness. These include worries and fear of infecting others, shame or stigma associated with being infected with COVID-19,[37] and uncertainty of the prognosis.[18] Indeed, the diagnosis of potentially life-threatening diseases, for example, cancer, has previously been associated with a dramatic rise in risks of psychiatric disorders[38] and suicide.[39] In addition, dysregulated immune response, for example, the documented cytokine storm associated with COVID-19,[40] may independently affect risks of psychiatric disorders in patients with COVID-19,[19] as has been documented in patients with other infections.[41] In line with Wang et al,[42] we found that individuals with a higher educational level are more likely than those with a lower educational level to suffer from symptoms of depression and anxiety after a COVID-19 diagnosis. These findings are intriguing and require further investigation as previous studies have also provided contradicting results.[43]

Our mediation analysis suggests that influenza-like symptom burden mediated the risk elevations in psychiatric morbidities among patients recovering from COVID-19. Moreover, influenza-like symptoms among individuals without a confirmed COVID-19 diagnosis

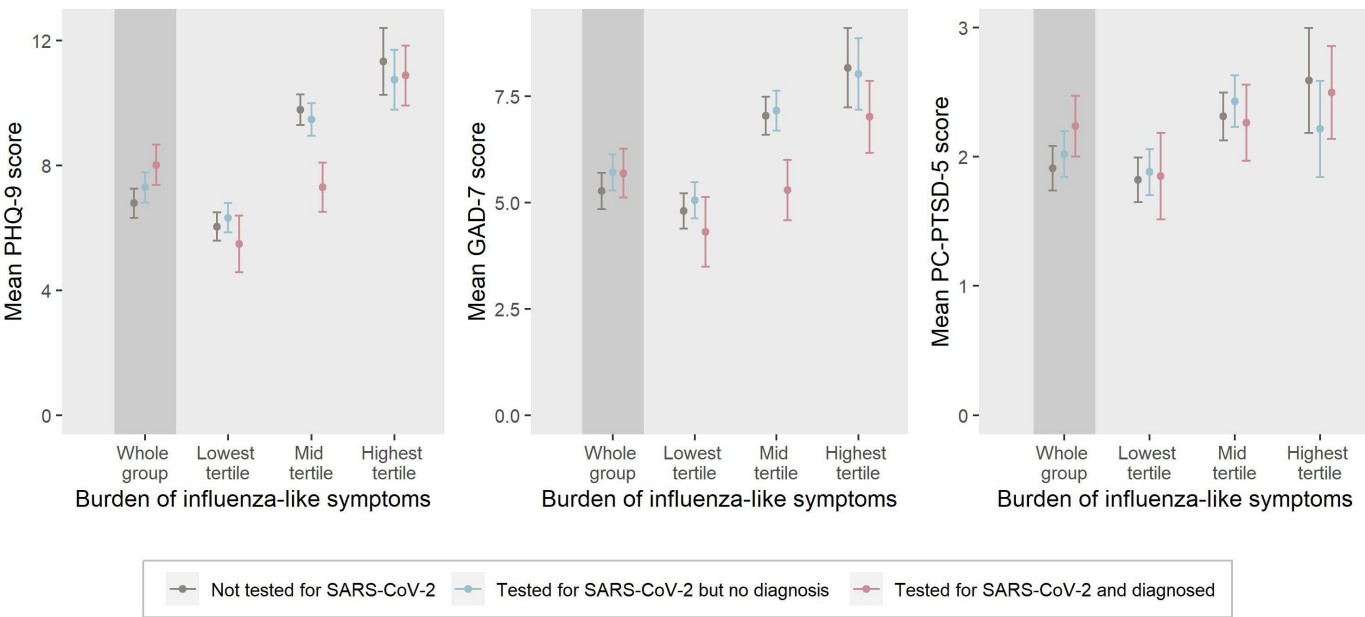

**Figure 2** Adjusted symptom scores of depression, anxiety and posttraumatic stress disorder (PTSD) by influenza-like symptom burden among individuals with a recent COVID-19 diagnosis, those who tested negative and those never tested for SARS-CoV-2.

were also strongly associated with symptoms of mental morbidities, suggesting that the association is not limited to COVID-19. Indeed, patients with a COVID-19 diagnosis reporting low–medium influenza-like symptom burden had lower levels of depression and anxiety than individuals without a diagnosis of COVID-19 but with similar levels of influenza-like symptoms. While the healthy prepandemic profile of the COVID-19 population may be an indicator of psychological resilience during recovery from COVID-19, it is also possible that the systematic, daily surveillance calls to all COVID-19 patients from the specialised COVID-19 outpatient clinic at the National University Hospital had a positive mental health impact on patients.

This population-based study includes more than 20% of all patients diagnosed with COVID-19 by RT-PCR during the first wave of the epidemic in Iceland. The diagnostic rate in Iceland was among the highest worldwide during the first wave of the epidemic,[44 45] allowing us to perform sensitivity analyses limited to individuals who all had been screened for SARS-CoV-2. The extensive and well-piloted questionnaire allowed us to obtain validated symptom assessment of mental morbidities along with detailed information on influenza-like symptom burden for all individuals during the influenza season in Iceland, together with a wide range of potential confounders in the association between COVID-19 and mental morbidities.

However, our study is cross-sectional and, therefore, less informative on the directionality of the studied associations. Indeed, a newly published study using data from the UK Biobank suggests that individuals with a history of prepandemic psychiatric disorders are more likely to be hospitalised and die from COVID-19.[46] Thus, it is possible that patients who were already experiencing

psychological symptoms suffered the most severe COVID-19-related symptomology yielding the observed associations. However, we included previous diagnoses of psychiatric disorders in all our models which should, to some extent, alleviate this concern. Nevertheless, it is possible that individuals with severe symptoms of depression and anxiety were more likely to report high levels of influenza-like symptoms during the preceding 2 months when responding to the questionnaire, yielding a detrimental outcome-dependent misclassification. Indeed, the instrument to assess influenza-like symptom burden included similar items (eg, fatigue) as are included in the PHQ-9 instrument which could, to some extent, explain the association between COVID-19 and depression. Yet, more objective assessments of illness severity, for example, the number of days confined to bed or hospitalised due to COVID-19—both of which were strongly associated with mental morbidities among patients—might to a lesser extent be influenced by the participants' mental status.

Participants recovering from COVID-19 may differ in terms of their mental health status from patients diagnosed with COVID-19 who did not participate in our study. While we were not able to disentangle the direction of such selection, our observed prevalence of depression and anxiety among patients recovering from COVID-19 resembles prevalence rates recently reported in a meta-analysis of diverse COVID-19 patient populations elsewhere.[9] Finally, our study is nested within a small, economically and socially secure society with virtually free and high-quality healthcare for all, thus, the findings may not be generalisable to other settings.

In conclusion, the relatively healthy first-wave COVID-19 population in Iceland presented with increased risk of depression and PTSD in the early weeks of recovery

from their illness, particularly those recovering from a severe disease. These findings motivate further follow-up studies of mental health among patients recovering from COVID-19 and other serious infections and heightened clinical surveillance of those recovering from a severe illness.

**Author affiliations**
[1]Centre of Public Health Sciences, Faculty of Medicine, University of Iceland, School of Health Sciences, Reykjavik, Iceland
[2]Landspitali–The National University Hospital of Iceland, Reykjavik, Iceland
[3]Faculty of Psychology, University of Iceland School of Health Sciences, Reykjavik, Iceland
[4]Faculty of Medicine, University of Iceland School of Health Sciences, Reykjavik, Iceland
[5]Institute for Sustainability Studies, University of Iceland, Reykjavik, Iceland
[6]The Directorate of Health, Reykjavik, Iceland
[7]Icelandic Heart Association, Kopavogur, Iceland
[8]Department of Medical Epidemiology and Biostatistics, Karolinska Institute, Stockholm, Sweden
[9]Department of Epidemiology, Harvard TH Chan School of Public Health, Boston, Massachusetts, USA

**Contributors** UV, TA and JJ had full access to all of the data in the study and take responsibility for the integrity of the data and the accuracy of the data analysis. UV, AH, KSS, IM, JJ, TA conceived and design the study. UV, AH, KSS, IM, JJ, TA analysed the data, and all authors (KSS, HÝH, IM, AH, EBT, ÁBG, GT, HR, HLJ, BG, GP, PHP, SYK, TJL, SH, HH, GG, EE, DGG, HS, SH, ADM, RP, JJ, TA, UV) interpreted the data. UV, KSS, HÝH, IM drafted the manuscript. All authors provided critical revision of the manuscript for important intellectual content. UV, AH, HR, GT, TA, JJ, SYK, TJL, SH, DGG, ADM provided administrative, technical, or material support. The corresponding author attests that all listed authors meet authorship criteria and that no others meeting the criteria have been omitted.

**Funding** This study was supported by grants from the Icelandic government and NordForsk (Mental morbidity trajectories in COVID-19 across risk populations of five nations, grant 105668, Dr. UV). The funding sources had no role in the design and implementation of the study; in data collection, management, analysis, and interpretation of the data; preparation, review, or approval of the manuscript; and decision to submit the manuscript for publication.

**Competing interests** None declared.

**Patient consent for publication** Not required.

**Ethics approval** The study was approved by the National Bioethics Committee of Iceland, Reykjavik (NBC no. 20-073) and the Data Protection Authority.

**Provenance and peer review** Not commissioned; externally peer reviewed.

**Data availability statement** The original data and R codes are available on request upon approval of the National Bioethics Committee and the Data Protection Authority.

**Author note** The lead author/guarantor of the study (UV) affirms that this manuscript is an honest, accurate and transparent account of the study being reported; that no important aspects of the study have been omitted; and that any discrepancies from the study as planned (and, if relevant, registered) have been explained.

**ORCID iDs**
Hildur Ýr Hilmarsdóttir http://orcid.org/0000-0002-9439-3022
Thor Aspelund http://orcid.org/0000-0002-7998-5433
Unnur Valdimarsdottir http://orcid.org/0000-0001-5382-946X

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
