## [Reviewer comments · BMJ Open]

ARTICLE DETAILS

TITLE (PROVISIONAL)	Illness severity and risk of mental morbidities among patients recovering from COVID-19: a cross-sectional study in the Icelandic population
AUTHORS	Sævarsdóttir, Karen; Hilmarsdóttir, Hildur; Magnúsdóttir, Ingibjörg; Hauksdóttir, Arna; Thordardóttir, Edda; Guðjónsdóttir, Ásdís; Tomasson, Gunnar; Rúnarsdóttir, Harpa; Jónsdóttir, Harpa; Guðmundsdóttir, Berglind; Pétursdóttir, Guðrún; Petersen, Pétur; Kristinsson, Sigurður; Love, Thorvardur; Hansdóttir, Sif; Harðardóttir, Hrönn; Gudmundsson, Gunnar; Eythorsson, Elias; Guðmundsdóttir, Dóra; Sigbjörnsdóttir, Hildur; Haraldsdóttir, Sigríður; Möller, Alma; Palsson, Runolfur; Jakobsdóttir, Jóhanna; Aspelund, Thor; Valdimarsdóttir, Unnur

VERSION 1 – REVIEW

REVIEWER	Trautmann, Sebastian Technische Universität Dresden
REVIEW RETURNED	25-Apr-2021

GENERAL COMMENTS	This is well-written manuscript describing a study on the association between COVID-19 illness severity and mental morbidities. The methods are sound and clearly described. I have only a few suggestions: 1. The introduction completely lacks a theoretical background. Although this is provided in greater detail in the discussion, I suggest to very briefly describe a theoretical model how illness severity could influence the mental health indicators considered, but also the other way around (e.g. immune system alterations in mental disorders) already in the introduction to provide a suitable background.2. The authors claim that this is a population-based study. Participants seem to stem from a number of different studies, including recruitment via social media. Is the final sample actually representative of the reference population? And is there a possibility that subject participated in more than one of the included studies and were therefore counted more than once?3. What was the distribution of the flu-like symptom sum score, why did the author decide to reduce this information to 3 groups and were these groups actually the best way to capture difference in this sum score?4. I think by just reporting associations adjusted for several baseline measures the authors made their analysis less informative than it could be. In addition of holding these measures constant, it would be interesting to know if the investigated associations vary depending on these measures (i.e. moderator
--

	analysis). These analyses would allow much more conclusions about vulnerable groups. 5. The interpretation of the stratified analyses seems difficult because the number of individuals diagnosed with COVID is very different and very small in some age groups. To show that associations actually differ between these groups, one would, again, have to conduct moderator analyses.
--	--

REVIEWER	Lee, Brian K. Drexel Univ, Epidemiology and Biostatistics
REVIEW RETURNED	12-May-2021

GENERAL COMMENTS	Statistical review The statistical analyses performed by Saevarsdottir et al. are well-designed and executed. I appreciate the thoughtful sensitivity analyses that address important issues. One additional sensitivity analysis I would suggest is to examine whether the questionnaire answer date is an effect modifier, eg via stratified analysis. The questionnaire answer date is perhaps the starkest difference between cases and non-cases (38% of cases answered May 23-July 22 vs. 11% of non-cases). This may potentially influence findings beyond simply being a potential confounder, if mental health changes over time, e.g. if mental health and stress changes with the epidemic and with the lovely Icelandic summer.
---

VERSION 1 – AUTHOR RESPONSE

REVIEWER COMMENTS:

Reviewer: 1

Dr. Sebastian Trautmann, Technische Universitat Dresden

Comments to the Author:

This is a very well-written manuscript describing a study on the association between COVID-19 illness severity and mental morbidities. The methods are sound and clearly described.

Authors' response: Thank you for the positive comments on our work.

1. The introduction completely lacks a theoretical background. Although this is provided in greater detail in the discussion, I suggest to very briefly describe a theoretical model how illness severity could influence the mental health indicators considered, but also the other way around (e.g. immune system alterations in mental disorders) already in the introduction to provide a suitable background.

Authors' response: Thank you for this comment. Suggested changes have been made.

'Introduction' section (page 6, line 132-137): There are several mechanisms through which COVID-19 may be associated with mental morbidity. Firstly, uncertainty regarding the disease course may give rise to symptoms of anxiety and depression in infected patients.[16] Secondly, serious infection may induce a cytokine storm,[17, 18] leading to exacerbation of the illness and development of psychological symptoms such as depression.[17] The reverse may also be true that psychological symptoms may cause more severe COVID-19 illness through excessive or dysregulated inflammation.[19]

2. The authors claim that this is a population-based study. Participants seem to stem from a number of different studies, including recruitment via social media. Is the final sample actually representative of the reference population? And is there a possibility that subject participated in more than one of the included studies and were therefore counted more than once?

Authors' response: Whilst we do agree that the analytic sample is not representative of the Icelandic population (as 70% of participants are women and the median age is somewhat higher than in the general population, see also Table 1), the questionnaire was open to all Icelanders who had a national identification number, and we did not exclude any participants. The study was widely advertised in various media in Iceland. It was also endorsed by the Directorate of Health. Furthermore, the ongoing prospective cohort studies in Iceland that permitted us to contact and invite their participants to answer our questionnaire were themselves population-based. We enriched our dataset by also contacting those individuals who were RT-PCR positive for SARS-CoV-2. Though it is possible that individuals who received an email invitation to participate in our study due to their involvement with one of the ongoing prospective cohort studies in Iceland may have participated in more than one of those cohorts, this would not have affected our study as each participant received an electronic ID that was linked to their national identification number which only allowed them to answer once.

We have now clarified how our analytic sample resembles and differs from the general population.

'Results' section (page 12, line 286-288): The analytic sample had similar education[33] and residence distribution as the general population, while the median age was higher and women were overrepresented.[34]

3. What was the distribution of the flu-like symptom sum score, why did the author decide to reduce this information to 3 groups and were these groups actually the best way to capture difference in this sum score?

Authors' response: The questions on the nine flu-like symptoms were designed for the purpose of this study in the middle of the first COVID-19 pandemic wave. The association between flu-like symptoms and mental health indicators were not completely linear, as can be seen in the attached figure (Title: flu_symptoms_vs_meanPHQ9_score.png) on flu and depressive symptoms. We therefore chose to divide sum scores into tertiles and present the results in that way. We are of course willing to alter our strategy on the editor's request, but we have confirmed that this approach will neither alter our results nor the conclusion.

4. I think by just reporting associations adjusted for several baseline measures the authors made their analysis less informative than it could be. In addition of holding these measures constant, it would be interesting to know if the investigated associations vary depending on these measures (i.e. moderator analysis). These analyses would allow much more conclusions about vulnerable groups.

Authors' response: Thank you for this comment. Please observe that we always report crude associations and then multivariable adjusted in all analyses. We have now made additional analysis, as requested (see supplementary Table 2), where we stratify for all covariates, one-by-one and perform an interaction test (moderator analysis). The results show that there is a significant effect modification by age for all three outcomes, by education for depression and anxiety, and by gender for PTSD and anxiety.

We have now added these results in supplement and referenced in text where appropriate.

'Abstract' section, Results (page 4, line 82): Elevated relative risks were limited to recovering COVID-19 patients 40 years or older and were particularly high among individuals with university education.

'Methods' section, Statistical analysis (page 10, line 239-240): We then performed these analyses stratified by all covariates and performed a likelihood ratio test for effect modification.

'Results' section (page 14, line 305-307): Individuals with higher educational level were more likely to suffer from symptoms of depression and anxiety after a diagnosis of COVID-19.

'Discussion' section (page 18, line 393-394): We found that mental morbidities among recovering COVID-19 patients were strongly associated with older age, higher educational level, greater flu-like symptom burden, extended time confined to bed and hospitalization due to COVID-19.

'Discussion' section (page 18, line 407-411): In line with Wang et al.,[43] we found that individuals with a higher educational level are more likely than those with a lower educational level to suffer from symptoms of depression and anxiety after a COVID-19 diagnosis. These findings are intriguing and require further investigation as previous studies have also provided contradicting results.[44]

5. The interpretation of the stratified analyses seems difficult because the number of individuals diagnosed with COVID is very different and very small in some age groups. To show that associations actually differ between these groups, one would, again, have to conduct moderator analyses.

Authors' response: In the manuscript, we only conclude that the associations between COVID-19 diagnosis and mental morbidity indicators were not observed in the youngest age group (18-39 years). In addition, according to the reviewer's suggestion, we now show that there is a significant effect modification by age (supplementary Table 2).

Reviewer: 2
Dr. Brian K. Lee, Drexel Univ

Comments to the Author:
Statistical review

The statistical analysis performed by Saevarsdottir et al. are well-designed and executed. I appreciate the thoughtful sensitivity analyses that address important issues.

Authors' response: Thank you for the positive comments on our study.

One additional sensitivity analysis I would suggest is to examine whether the questionnaire answer date is an effect modifier, eg via stratified analysis. The questionnaire answer date is perhaps the starkest difference between cases and non-cases (38% of cases answered May 23-July 22 vs. 11% of non-cases). This may potentially influence findings beyond simply being a potential confounder, if mental health changes over time, e.g. if mental health and stress changes with the epidemic and with the lovely Icelandic summer.

Authors' response: Thank you for this comment. We have made an additional stratified analysis as suggested. The results suggest somewhat higher relative risk of depression and anxiety among COVID-19 patients answering at the start of the data collection. This may be due to a relatively

shorter time from the COVID-19 diagnosis compared to those answering later. Higher relative risks were observed for PTSD among individuals with COVID-19 in the latter response period, and no statistically significant interactions were observed. We now report the results of this analysis in supplement and have made corresponding changes to the manuscript.

'Methods' section, Statistical analysis (page 10-11, line 246-250): There was a considerable difference between individuals with and without a COVID-19 diagnosis with respect to the date of responding to the questionnaire (Table 1). Therefore, we repeated the multivariable Poisson regression stratified by questionnaire answer date, divided into two groups (April 24-May 7, and May 8-July 22).

'Results' section, COVID-19 and mental morbidities (page 14, line 307-310): When stratified by answer date, risk elevations for depression and anxiety in COVID-19 patients were lower for those who answered in the later response period (i.e., May 8-July 22), whereas it was higher for PTSD (see supplementary Table 2) and no statistically significant interactions were observed.

VERSION 2 – REVIEW

REVIEWER	Trautmann, Sebastian Technische Universität Dresden
REVIEW RETURNED	01-Jul-2021
GENERAL COMMENTS	The authors carefully addressed all comments raised by the reviewers. I have no further suggestions.